# An Efficient Convolutional Neural Network Model Combined with Attention Mechanism for Inverse Halftoning

**Linhao Shao** [1], **Erhu Zhang** [1,2,*] **and Mei Li** [3]

1   School of Printing, Packaging and Digital Media Technology, Xi'an University of Technology,
    Xi'an 710048, China; lh_shao2101@163.com
2   Department of information Science, Xi'an University of Technology, Xi'an 710048, China
3   School of Mechanical and Precision Instrument Engineering, Xi'an University of Technology,
    Xi'an 710048, China; limei_ys@foxmail.com
*   Correspondence: eh-zhang@xaut.edu.cn; Tel.: +86-029-82312435

**Abstract:** Inverse halftoning acting as a special image restoration problem is an ill-posed problem. Although it has been studied in the last several decades, the existing solutions can't restore fine details and texture accurately from halftone images. Recently, the attention mechanism has shown its powerful effects in many fields, such as image processing, pattern recognition and computer vision. However, it has not yet been used in inverse halftoning. To better solve the problem of detail restoration of inverse halftoning, this paper proposes a simple yet effective deep learning model combined with the attention mechanism, which can better guide the network to remove noise dot-patterns and restore image details, and improve the network adaptation ability. The whole model is designed in an end-to-end manner, including feature extraction stage and reconstruction stage. In the feature extraction stage, halftone image features are extracted and halftone noises are removed. The reconstruction stage is employed to restore continuous-tone images by fusing the feature information extracted in the first stage and the output of the residual channel attention block. In this stage, the attention block is firstly introduced to the field of inverse halftoning, which can make the network focus on informative features and further enhance the discriminative ability of the network. In addition, a multi-stage loss function is proposed to accelerate the network optimization, which is conducive to better reconstruction of the global image. To demonstrate the generalization performance of the network for different types of halftone images, the experiment results confirm that the network can restore six different types of halftone image well. Furthermore, experimental results show that our method outperforms the state-of-the-art methods, especially in the restoration of details and textures.

**Keywords:** inverse halftoning; convolutional neural networks; attention mechanism; multi-stage loss function





## 1. Introduction

Digital halftone is a technique to convert a continuous-tone image into a binary image known by the name of halftone image. Due to the low pass character of the human eyes, the generated halftone image can be perceived as a continuous-tone image when viewed from a certain distance. Thus, the digital halftone technique is widely used in bi-level output devices in order to reproduce the tone of a continuous-tone image, such as printing press machines, printers, fax machines and so on [1,2]. Besides, the digital halftone technique can also be employed as an image compression mode for saving storage space or electric power in special occasions, for example in telemedicine [3] and IoT [4]. Major halftone methods used in practice include ordered dithering (OD), dot diffusion (DD), error diffusion (ED) and direct binary search (DBS) [5].

Inverse halftoning is the reverse process of digital halftone, which is used to restore a continuous-tone image from its halftone version. The reason of using inverse halftoning is

that the typical image processing approaches, such as zooming, rotation, compression or feature extraction, etc., cannot be directly applied to halftone images. When we want to further reuse the halftone images appearing in newspapers, magazines or books, they have to be firstly restored as the corresponding continuous-tone images by using inverse halftoning technique. There are many important applications of inverse halftoning. First, effective inverse halftoning can be used to restore images that are compressed with lower bits. In addition, the photographs of halftone pattern in books and magazines can be scanned and transformed into continuous tone image, which is meaningful for that historically important photos on old newspapers. Furthermore, continuous-tone images need to be reconstructed for image enhancement, image manipulations or image super resolution to obtain better results. Due to the inevitable information losses in the process of digital halftone, inverse halftoning is an ill-posed problem which means there is not a unique solution for this problem. In addition, noise dot-patterns are often blended into halftone images. Hence, inverse halftoning is a challenging research area.

Since 1990s, many inverse halftoning methods have been proposed. These presented methods can be roughly divided into traditional methods and deep learning-based methods [6]. The traditional methods include filters [7,8], projection onto convex sets method (POCS) [9], maximum a posteriori (MAP) estimation method [10], wavelet-based method [11], look-up-table method (LUT) [12,13], dictionary learning method [2,14], and neural networks method [15]. Although the above traditional approaches have all achieved excellent performance at the time, the restored continuous-tone images still suffer several from visual artifacts and subtle details loss. Deep learning, in particular deep convolutional neural networks (DCNN), have shown their powerful performance for many computer vision applications. However, there are few studies in the field of inverse halftoning using DCNN. Considering inverse halftoning as an image transformation problem, the first application of DCNN for inverse halftoning was studied by Hou and Qiu [16], who demonstrated that the method based on deep learning is superior to the traditional methods. Afterwards, more and more studies were focused on inverse halftoning based on deep learning [17–22]. According to the network architecture, three typical architectures were explored in the above proposed methods, which are U-Net [16,19], residual network [18,20] and fully convolutional networks [17]. In terms of the type of halftone images, inversed halftoning methods can be classified into methods for digital halftone images and methods for scanned halftone images. Nowadays, most of the existing methods [16–20] address digital halftone images, and only a few of methods focus on scanned halftone images [21,22]. Although these methods have made a number of contributions to inverse halftoning, there are still problems in this field, such as improving the fine image details and increasing the model adaptation ability to different types of halftone images.

The critical problem of inverse halftoning is to remove noise dots on flat areas and restore image details on textured areas [17]. However, removing noise dots and restoring image details form a pair of contradictions. As a result, the first problem of existing methods based on deep learning is not well removing halftone noise dots, while the second problem is that the subtle details need to be further improved in the restored image. In addition, due to the diverse distributions of black dots and white dots in different types of halftone images, the adaptation ability of the existing methods is poor to different types of halftone images. To address these issues, inspired by the successful applications of attention mechanism in many image restoration tasks, we propose a new inverse halftoning method by incorporating attention mechanism into DCNN, which can determine where to focus and suppress. Recently, attention mechanisms have been an important component in deep learning. Among the many diversifications of attention module, the residual channel attention block (RCAB) [23] showed effectiveness in super-resolution [24], image denoising [25] and so on. Therefore, we employ RCAB in the proposed network, which can guide the network to pay more attention on essential channel features and suppresses unnecessary ones. Moreover, the attention mechanism can increase the network adaptation ability to restore different types of halftone images. Particularly, [26] proposed that the

different layers of CNN contain different feature information. The low-level features contain more sharp and detailed information, while high-level contain more abstract semantic information. So, in this article, the low-level detail information and high-level semantic information are concatenated with skip connection and then fused by attention blocks, which helps to restore image details. To further improve the restored image fine details, multi-stage loss functions are proposed in the presented network. The final loss function integrates multiple loss functions which come from the restored images at different stages, this helps to obtain more informative features at different restoring stages and thus improve the restored image fine details. We conduct extensive experiments on VOC2012 dataset [27] and six types of halftone images. Results show that our method outperforms the state-of-the- art methods and has better generalization performance for different types of halftone images.

The contributions of the paper are as follows:

(1) We introduce the attention mechanism to the proposed network, which can better guide the network to remove noise dot-patterns and restore image details, and improve the network adaptation ability. To the best of our knowledge, this is the first work of using attention mechanism for inverse halftoning.

(2) Multi-stage loss functions are employed in the network, which can further enhance the restored image details.

(3) The experimental results demonstrate that the proposed method achieves impressive performances compared with the state-of-the-art methods and can be applied to many different types of halftone images.

The rest of the paper is organized as follows: Section 2 briefly introduces the related works about inverse halftoning including traditional inverse halftoning methods and deep learning-based methods. Section 3 details the proposed method. Section 4 discusses the experimental results. Finally, Section 5 concludes the paper.

## 2. Related Works

As a classic image restoration problem, inverse halftoning has been widely developed in the past decades, and a large number of inverse halftoning methods have been proposed, including filtering methods [8], wavelets [11], look-up table (LUT) [12] and neural networks [16–18,20]. Especially in recent years, with the rapid development of deep learning, supervised learning and unsupervised learning have been widely used in industrial production [28–30], which promotes the rapid development of inverse halftoning. As our method belongs to deep learning method, we review the progresses in inverse halftoning according to the traditional inverse halftoning methods and deep learning based inverse halftoning methods.

### 2.1. Traditional Inverse Halftoning

In view of the characteristics of low pass filtering of human eyes, and high frequency characteristics of halftone dot-patterns, the earliest methods were based on low filters, such as the Gaussian filter, median filter or bilateral filter [31]. Low pass filtering simply removes halftone dot-patterns, but image details are also removed by this process. To get more image details in the restored images, adaptive filtering [8], non-linear filtering [32] and transform-domain filter [11,33] were investigated. Kite et al. [8] proposed a multi-scale gradient estimation filter, and then used it to choose the best parameterized smoothing filter from a family of parameterized customized smoothing filters for each pixel. The proposed method can obtain a sharp image with a low perceived noise level for error- diffused halftone images. Kim et al. [32] proposed a non-linear binary permutation filter for reconstructing continuous-tone images from ordered dithered halftone or error diffused halftone images. The presented filter is based on the space and rank orderings of the halftone samples in a halftone observation window. Luo et al. [33] proposed a novel wavelet-based inverse halftoning method, which can remove halftone noise by noise attenuation and intraband filtering in the wavelet space. The method presented by Xiong et al. [11] used highpass

wavelet images and cross-scale correlations in the multiscale wavelet decomposition to remove halftone noise while preserving image edges and details information. Unlike the aforementioned filtering method, Mese et al. [12] creatively proposed a look up table (LUT) inverse halftoning method with fast speed. In this method, the LUT was constructed by a halftone template and its corresponding continuous-tone value. Inspired by the successful applications of sparse representation in field of signal processing, a novel inverse halftoning method based on sparse representation was firstly presented by Son [34], where two jointed dictionaries are learned for the concatenated feature spaces of continuous-tone images and halftone images. In the method, there is assuming that the sparse representation coefficients of continuous-tone images and halftone images are the same. To relax the assumption, Zhang et al. [2] proposed a semi-coupled multi-dictionary learning method for inverse halftoning. Son [14] proposed an edge-oriented local learned dictionaries (LLD) method, which can enhance the edge details of the restored image. Considering the quality of inverse halftoning depends on the starting halftone method, Huang et al. [15] proposed an inverse halftoning method based on neural network by integrating the process of digital halftoning and inverse halftoning, where a single-layer perceptron neural network was adopted for halftoning and a radial-basis function neural network was adopted for inverse-halftoning. Although these abovementioned methods produced relatively satisfactory results at that time, the quality of restored image by these conventional methods is still not as good as those based on deep learning methods. The reason is that the method based on deep learning can get deep and hierarchical feature representation in an end to end manner, which is more efficient to extract abstract features for halftone image restoration. Moreover, features extracted at different levels based on deep learning method exhibit diverse characteristics to the input halftone image. Thus, the method based on deep learning can better restore image details by fusing low-level detail features and high-level semantic features. In references [16–18,20], they compared with some classical traditional methods, such as filtering method [8], wavelet method [11], LUT method [12], MLP method [15] and dictionary learning method [14]. The experimental results also demonstrate the image quality restored by inverse halftoning methods based on deep learning is superior to traditional inverse halftone methods both in qualitative and quantitative evaluation aspects.

*2.2. Deep Learning Based Inverse Halftoning*

Deep convolutional neural networks have shown their outstanding performance for many tasks. Hou and Qiu [16] firstly applied DCNN to inverse halftoning, where they used a U-net network as the transformation network for inverse halftoning. In addition, perceptual loss based on pre-trained network was also introduced to construct the objective function for the training, which can overcome the shortcoming of per-pixel loss of producing blurry outputs. To obtain more image details, Xiao et al. [19] proposed a two- stages gradient-guided DCNN for inverse halftoning. In the first stage, two subnetworks are designed to predict the gradient maps from the input halftone image. In the second stage, the gradient maps, along with input halftone image, are fed to the third subnetwork to reconstruct the continuous-tone image. All the three subnetworks are the U-net architecture, and halftone images generated by the Floyd-Steinberg error diffusion algorithm are used to perform experiments. On the basis of the method [19], Yuan et al. [20] put forwarded a gradient-guided residual learning method for inverse halftoning. In the paper, the second stage is a residual network, which helps to restore better local details. Xia et al. [18] proposed deep inverse halftoning via progressively residual learning (PRL), which is another foundational work for inverse halftoning. The PRL includes two modules: content aggregation module that is used to remove halftone noise and reconstruct the initial continuous-tone image, and detail enhancement module that extracts fine structures by learning a residual image. Recently, Son [17] presented a structure-aware DCNN (SADCNN) for inverse halftoning, which can not only remove noisy dot-patterns well on flat areas but also restore details clearly on textured areas. Guo et al. [35] firstly pro-

posed a novel inverse halftoning method by using GAN network, which can effectively perform both halftoning and inverse halftoning for dispersed dot halftone images. Due to no paired data of halftone images and their corresponding continuous-tone images, restoring scanned halftone image is more challenging than restoring digital halftone image. Kim T.H et al. [22] proposed a context-aware descreening method for scanned halftone image. The method consists of two main stages, where the intrinsic features of the scene are extracted for reconstructing the low-frequency of the image and removing halftone noise at the first stage, and fine details are synthesized on top of the low-frequency output at the second stage. Gao et al. [21] proposed a novel inverse halftoning method for scanned halftone images. In the method, the first stage is unsupervised training for removing printing artifacts which make the method adapt to real halftone prints, and the second stage is a supervised training manner for the inverse of halftoning by using synthetic training data. Table 1 summarizes the traditional inverse halftoning methods and deep learning-based inverse halftoning methods.

**Table 1.** A comparative summary of related work.

| Methods | Representative Sub-Method | Pros | Cons |
|---|---|---|---|
| Traditional inverse halftoning | Filtering method | It can remove halftoning noise | It also removes edge information |
| | Wavelet method | It can preserve important edge information | Only for grayscale halftones |
| | Maximum a-posteriori (MAP) method | It can reconstruct the smooth regions of the image and the discontinuities image edges | The poor quality of the restored images |
| | Look-up table (LUT) method | Fast | Depend on the choice of table |
| Deep Learning Based Inverse Halftoning | U-net network architecture | The model is simple and efficient | The details are not well restored |
| | Gradient-guided residual learning method | Restore better local details | Restoration of single type image |
| | Progressively residual learning method | Remove halftone noise and detail enhancement | Generalization of restoration is not good |
| | Structure-aware DCNN method | Remove noisy dot-patterns and restore details clearly on textured areas | Only designed for grayscale halftones |

As can be seen from the abovementioned methods, we can conclude: (1) Most of the proposed methods are focused on error diffusion halftone images. However, there are more than twenty types of halftone images used in practice, which have different critical halftone characteristics such as dot directivity, dot distribution, pattern periodicity, directional artifacts and so on. [1,5]. Therefore, a general purpose inverse halftoning method which has generalization performance for different types of halftone images is urgently need. (2) Due to the coexistence of details loss and halftone noise, the restored images still suffer from fine details loss and visual artifacts. To solve these problems, we propose a novel inverse halftoning method in which the attention mechanism and multi-stage loss functions are introduced to better work with the proposed network. The proposed method is simple yet effective for different types of halftone images.

## 3. Methodology

In this study, we propose a novel inverse halftoning method based on deep learning, which integrates deep CNN, attention module and multi-stage loss functions. Firstly, we will introduce the network architecture. Then, the multi-stage loss functions will be discussed.

### 3.1. Network Architecture

As shown in Figure 1, the architecture of the proposed approach consists of three major components: (1) Feature extraction and halftone noise removing; (2) Image restoring with residual channel attention block (RCAB) module and contextual semantic information aggregation; (3) Multi-stage loss functions learning. The first part begins with a normal convolution. As is evidenced in reference [36], a large convolutional kernel could be replaced by a multi-layer convolution with small kernel size, which can reduce the parameter count and improve the non-linear ability of the network. To keep the size of the output feature maps as the same of the inputs, the stride and the padding have to be set as 1. Since initial convolution is used to extract low-level features, such as edges, corners, lines and colors, they can be used to restore image details in the second part with skip connection. Therefore, the number of filters in the initial convolution layer is set as 32 in order to extract more fine-grained features. Then, three convolution blocks (Conv_Block1, Conv_Block2 and Conv_Block3) are cascaded for further refining features and removing halftone noise, where each Conv_Block includes three sequential basic units: convolution, LeakyRelu and convolution. For each Conv_Block, the filter number of the first convolution is the same as the input channel number of this Conv_Block, and the filter number of the last convolution is twice of the first convolution. The above feature extraction process can be expressed in the following formula:

$$X = \text{Conv\_Block3}(\text{Conv\_Block2}(\text{Conv\_Block1}(\text{Conv}(X_I)))) \tag{1}$$

where $X_I$ is the input image and $X$ is the extracted feature image.

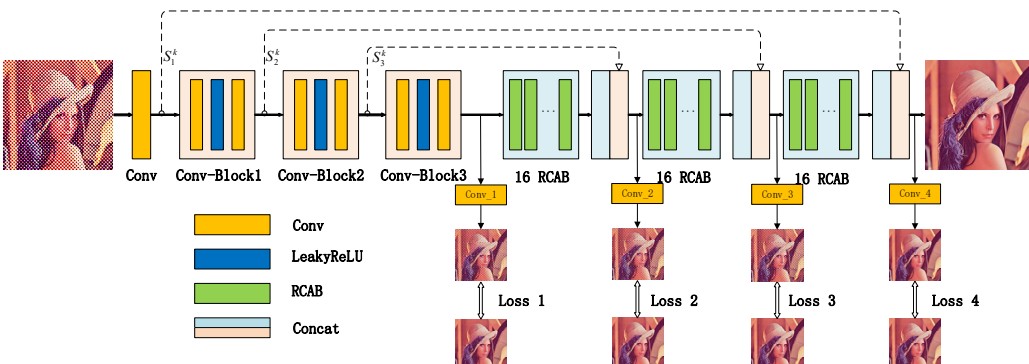

**Figure 1.** The proposed network architecture.

In the second part, three attention modules and three concatenation modules with identical layout are used to restore continuous-tone images. In every attention module, we sequentially lay 16 RCAB to extract the channel statistic among channels to further enhance the discriminative ability of the network [23]. The details of residual channel attention block are shown in Figure 2. As shown in Figure 2, $X_{b-1}$ is the input of RCAB block:

$$F_b = \omega_b^2 \delta\left(\omega_b^1 X_{b-1}\right) \tag{2}$$

where $\omega_b^1$ and $\omega_b^2$ are weight sets of the two convolutional layers in RCAB. $\delta(.)$ denote ReLU function. Let $F_b = [f_1, \ldots, f_c, \ldots, f_C]^{C \times H \times W}$ as input feature maps, which has C channels

and the size of feature map is $H \times W$. The channel-wise statistic of the c-th channel $z \in R^C$ is determined by Equation (3):

$$z_c = \frac{1}{H \times W} \sum_{i=1}^{H} \sum_{j=1}^{W} f_c(i, j) \tag{3}$$

where $(i, j)$ is the position of c-th channel $f_c$, $f_c(i, j)$ is the value at $(i, j)$. This formula denotes the global pooling for every channel, such channel statistic can express the whole image. The channel-wise global spatial information can be viewed as a channel descriptor by using global average pooling.

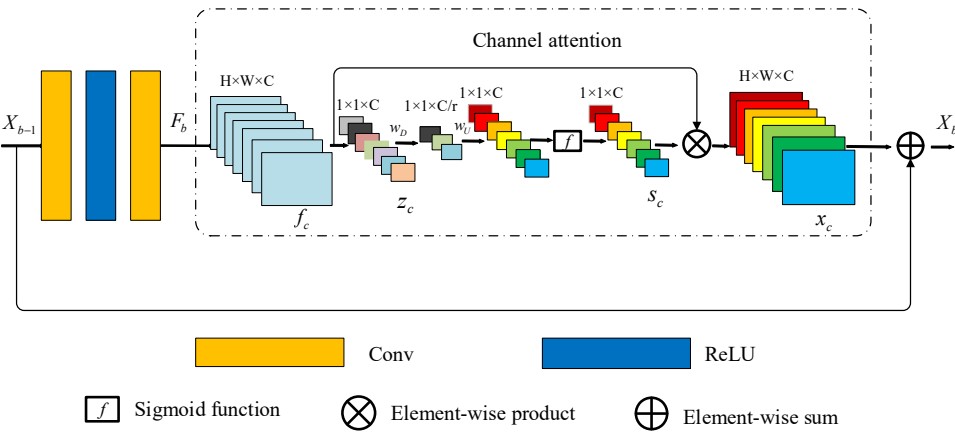

**Figure 2.** Residual channel attention block (RCAB).

We aggregate global information by average pooling. Then in order to capture channel-wise dependencies by introducing a gating mechanism. This gating mechanism can learn nonlinear interactions between channels and a non-mutually-exclusive relationship between channel-wise features. We opt to simple gating mechanism by sigmoid function:

$$s_c = f(\omega_U \delta(\omega_D z_c)) \tag{4}$$

where $\delta(.)$ and $f(.)$ denote ReLU function and the sigmoid gating, respectively. $\omega_D$ is the weight of a channel-downscaling layer with reduction ratio $r$. Then it is activated by ReLU. $\omega_U$ is the weight set of channel-upscaling with ratio $r$. Finally, the channel statistics $s_c$ is used to rescale the input feature map $f_c$:

$$x_c = s_c.f_c \tag{5}$$

where $s_c$ is the scaling factor in the c-th feature map. The channel attention enhances the discriminative ability by rescaling the residual component in the RCAB. For the b-th RCAB block, we have:

$$X_b = X_{b-1} + R_b(F_b) \tag{6}$$

where $X_{b-1}$ and $X_b$ are the input and output of residual channel attention block. $R_b$ dnotes channel attention module.

Following each attention module, there is a concatenation module which stacks the output of the attention module and the feature maps from the previous Conv_Block with skip connection. Before concatenating, the output of each attention module is convoluted with convolution in order to have the same channel number with the feature maps from the corresponding previous Conv_Block. The concatenation can provide more contextual semantic information, which helps to restore the fine details. The detailed parameters of the whole network are shown in Table 2.

**Table 2.** The detail experimental parameters of the network.

| | |
|---|---|
| **Feature extraction stage** | Stage1: [Conv], kernel_size = 3 × 3; C=16, 32; padding = 1 |
| | Stage2: Conv-Block1 [Conv + LeakyReLU + Conv] kernel_size = 3 × 3;C = 16, 32; padding = 1 |
| | Stage3: Conv-Block2 [Conv + LeakyReLU + Conv] kernel_size = 3 × 3;C = 32, 64; padding = 1 |
| | Stage4: Conv-Block3 [Conv + LeakyReLU + Conv] kernel_size = 3 × 3;C = 64, 128;padding = 1 |
| **Image reconstruction stage** | Stage1: 16 RCAB {[Conv + ReLU + Conv] kernel_size = 3 × 3; C = 128,128; padding = 1 [Avg_pool + Conv + ReLU + Conv+sigmoid] r = 16,kernel_size = 1 × 1; C = 8128; padding = 0} |
| | 1 × 1 Conv + Concat |
| | Stage2: 16 RCAB {[Conv +ReLU+Conv] kernel_size = 3 × 3; C = 128,128; padding = 1 [Avg_pool + Conv + ReLU + Conv+sigmoid] r = 16,kernel_size = 1 × 1; C = 8128; padding = 0} |
| | 1 × 1 Conv + Concat |
| | Stage3:16 RCAB {[Conv +ReLU+Conv] kernel_size = 3 × 3; C = 128,128; padding = 1 [Avg_pool + Conv + ReLU + Conv + sigmoid] r = 16, kernel_size = 1 × 1; C = 8,128; padding = 0} |
| | 1 × 1 Conv + Concat |

The whole network architecture imitates the design of U-net network structure, but our network doesn't adopt down-sampling and upsampling. Based on the analysis of U-net, we proposed the following special structure designs: (1) the downsampling can lead to the loss of underlying feature, which are lose some important feature information for the restoration of inverse halftoning. Thus, the downsampling is not adopted in the step of feature extracting in our proposed solution; (2) we use skip connection to retain detailed features from shallow layers, which can better restore the image details; (3) the special design of our network is the introduction of attention module, which can focus on informative features and further enhance the discriminative ability of the network. The ablation experiment results detailed in section IV demonstrate the rationality of attention module RCAB. (4) deep supervision with multi-stage losses can accelerate the optimization of network model. Unlike previous methods which compute loss function only using the final restoring image, we propose a multi-stage loss function calculating strategy. As shown in Figure 1, the loss functions of Loss1, Loss2, Loss3 and Loss4 are from images reconstructing at different stages, where Conv_1, Conv_2, Conv_3 and Conv_4 are used to reconstruct the continuous-tone images with three channels at different stage.

Through the analysis of the network architecture, we can conclude: (1) the proposed network is a fully convolution network, which can adapt to the input halftone images with different image size and can be learned in an end-to-end manner. (2) the proposed network is a lightweight network, where the parameter size is 10.87 M. (3) unlike the existing inverse halftoning method based on deep learning, which treat halftone image channel-wise features equally, the proposed method can generate different attention in different channel-wise feature. Thus, it is flexible to different types of halftone images and can pay more attention to the informative features at different channels, such as shapes, colors, edges and textures. (4) The proposed network is simple yet effective, because it achieves notable performance improvements compared to previous inverse halftoning methods.

*3.2. Loss Function*

We define multi-stage loss functions that measure the difference of the reconstruction image at different feature reconstruction stages and its original continuous-tone image. In feature reconstruction stage, In order to encourage the pixels of the construction image $\hat{y}$ to exactly match the continuous-tone image y, we hope them to have similar feature representations in different feature reconstruction layers.

As shown in Figure 3, when we reconstruct from early layers, the image content and global spatial structure are preserved. But when we reconstruct the continuous-tone image from higher layers, the color and texture are reconstructed well, so we define multi-stage loss functions to encourage the reconstruction image $\hat{y}$ to be perceptually similar to the continuous image at different feature reconstruction stages. What's more, if the output images of early layers are reconstructed well due to constraints of loss function, the reconstruction images of higher layers must be well.

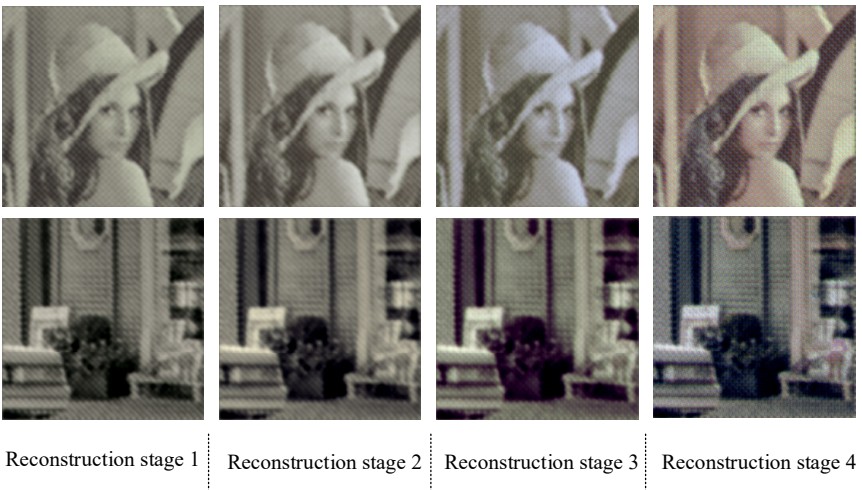

Reconstruction stage 1    Reconstruction stage 2    Reconstruction stage 3    Reconstruction stage 4

**Figure 3.** Illustration of the restoring images at different feature reconstruction layers.

Based on above analysis, as shown in Figure 1, we define multi-stage loss for training our network:

$$L(\hat{y}, y) = \sum_{j=1}^{4} \lambda_j \times \left| \hat{y}_j - y \right| \tag{7}$$

where $j$ denotes the j-th layer of reconstruction stage, $\hat{y}_j$ and $y$ are the output reconstruction image of j-th layer and the ground truth continuous-tone image. For the value of $\lambda$, we consider the following factors: In the deep layers, the network reconstructs good semantic information, the higher loss weight can better reconstruct the global information of the image. From deep layers to shallow layers, the loss weight from large to small is conducive to better reconstruction of the global image. By the experiment in Section 4, we set $\lambda_1 = 0.1$, $\lambda_2 = 0.2$, $\lambda_3 = 0.3$, $\lambda_4 = 0.4$.

## 4. Experiment

In this section, we introduce our experimental settings in Section 4.1, including detailed experimental parameters and dataset. Then we show the generalization performance of the proposed method for different types of halftone images in Section 4.2. To demonstrate the superior performance, we compare our method with the state-of-the art methods in Section 4.3. The ablative study in Section 4.4.

*4.1. Experiment Settings*

For training and testing, we use the PASCAL VOC2012 dataset [27], which are used as continuous-tone color images. The numbers of color images are 13,600, in which we select

10,000 images for training and 3600 images for validation. In addition, some other classical images are also selected as test images, such as Lena, Peppers, Baboon and so on.

Our network is implemented using the PyTorch frame-work and trained on 1080 Ti GPU. In the training, the whole network is trained in an end-to-end manner. We define the initial learning rate 0.0001 and choose Adam algorithm as the optimizer, the learning rate is reduced by Cosine annealing [37]. We select the number of iterations is 100 times and the batch size is 2 by experiments.

### 4.2. Evaluation for Different Types of Halftone Images

In order to illustrate the generalization performance of the network for different types of halftone image restoration. We choose different halftone methods to produce different types of halftone images. The continuous-tone images are converted into halftone images by Bayer's CD Ordered Dithering (BCD), Bayer's DD Ordered Dithering (BDD), Knuth DD Dot Diffusion (KDD), Direct Binary Search (DBS), Floyd Steinberg DD error diffusion (FSDD), Ulichney CD Ordered Dithering (UCD) respectively, the code is provided by Guo et al. [5]. Each type of halftone image is divided into training image and testing image.

For performance evaluation, the peak signal-to-noise ratio (PSNR) and the Structural Similarity (SSIM) are used to measure the difference of the output image and the original image. We train the network with different types of halftone images, and then twenty classical images are used to test the network. Figure 4 shows the classical images [38], numbered as 1–20.

The restoring continuous-tone images and their corresponding halftone images for Lena, Man and Peppers images are illustrated in Figure 5. From Figure 5, we can see the quality of the restoring images is satisfactory compared with the original images, which means the proposed method has generalization performance for different types of halftone images. In addition, the halftone images generated by DBS and FSDD have a great superiority over the other halftone images. Table 3 details the PSNR and SSIM values of different types of halftone images for the twenty classical images. From the average PSNR and SSIM, the proposed method achieves encouraging results on all types of halftone images.

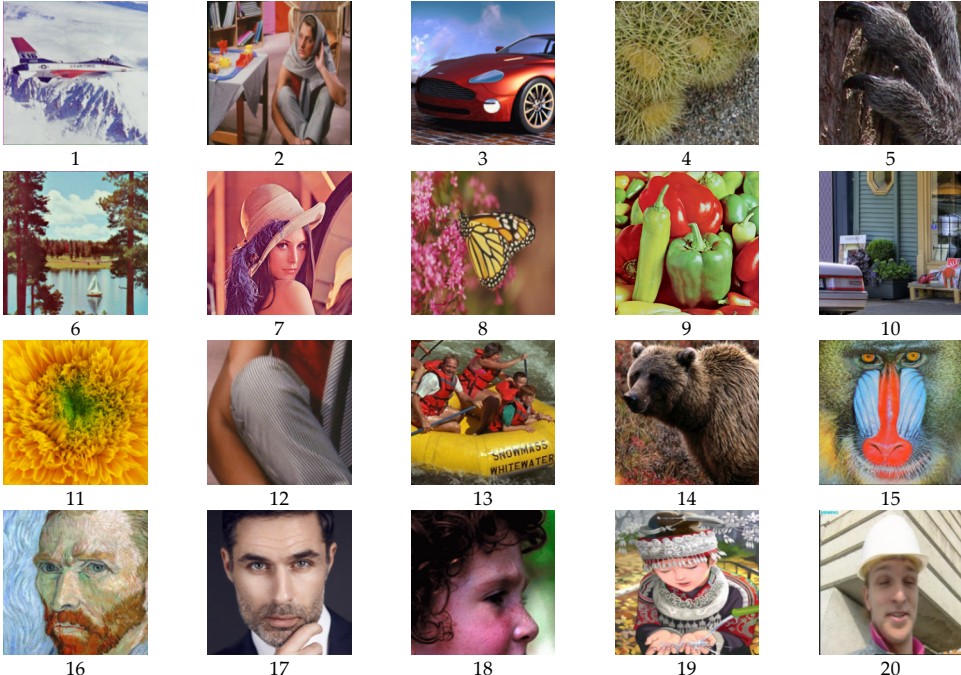

**Figure 4.** Several classical testing images.

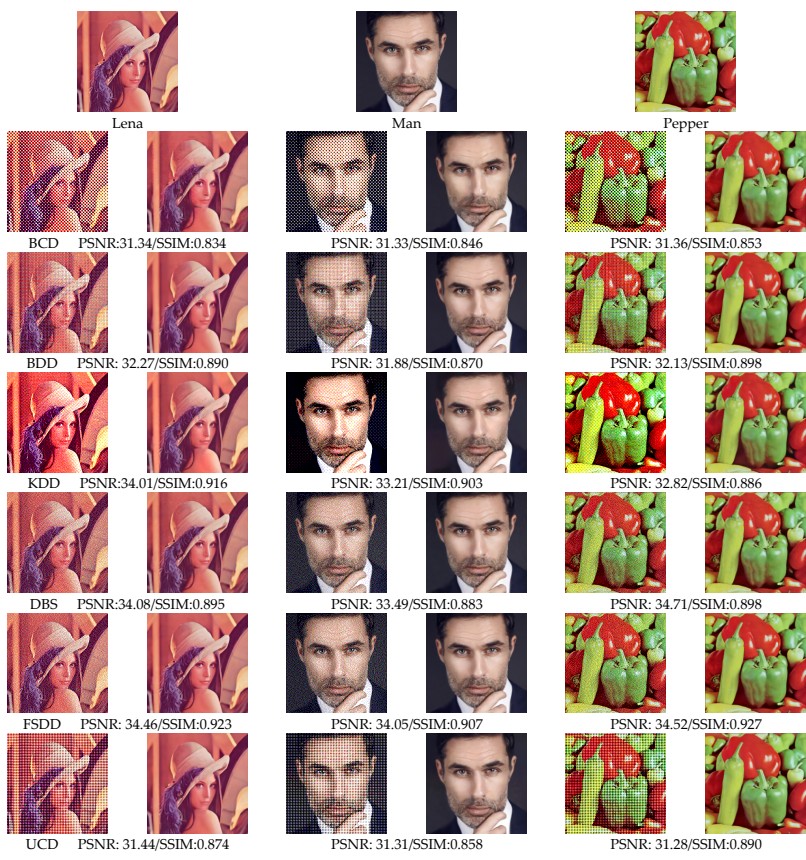

**Figure 5.** Experimental results of different types of halftone images for Lena, Man and Peppers images.

**Table 3.** Performance evaluation for different types of halftone image.

| Test | BCD | | BDD | | KDD | | DBS | | FSDD | | UCD | |
|---|---|---|---|---|---|---|---|---|---|---|---|---|
| | PSNR | SS IM | PSNR | SS IM | PSNR | SS IM | PSNR | SS IM | PSNR | SS IM | PSNR | SS IM |
| 1 | 28.2 | 0.85 | 28.9 | 0.88 | 32.7 | 0.93 | 31.9 | 0.92 | 32.2 | 0.93 | 28.0 | 0.87 |
| 2 | 29.1 | 0.83 | 28.8 | 0.86 | 32.8 | 0.92 | 30.8 | 0.86 | 31.9 | 0.92 | 29.0 | 0.85 |
| 3 | 30.1 | 0.86 | 30.9 | 0.89 | 30.9 | 0.85 | 33.2 | 0.88 | 33.4 | 0.92 | 29.9 | 0.87 |
| 4 | 24.1 | 0.68 | 25.6 | 0.80 | 29.3 | 0.93 | 28.1 | 0.86 | 28.9 | 0.91 | 24.8 | 0.75 |
| 5 | 24.1 | 0.65 | 25.1 | 0.72 | 27.9 | 0.80 | 27.6 | 0.83 | 28.8 | 0.88 | 24.4 | 0.68 |
| 6 | 26.8 | 0.79 | 27.7 | 0.84 | 31.0 | 0.89 | 30.5 | 0.88 | 30.9 | 0.91 | 27.2 | 0.83 |
| 7 | 31.3 | 0.83 | 32.3 | 0.89 | 34.0 | 0.92 | 34.1 | 0.89 | 34.5 | 0.92 | 31.4 | 0.87 |
| 8 | 28.6 | 0.87 | 29.9 | 0.92 | 32.9 | 0.95 | 32.5 | 0.93 | 33.2 | 0.96 | 29.3 | 0.91 |
| 9 | 31.4 | 0.85 | 32.1 | 0.89 | 32.8 | 0.89 | 34.7 | 0.89 | 34.5 | 0.93 | 31.3 | 0.89 |
| 10 | 26.2 | 0.78 | 26.6 | 0.82 | 29.0 | 0.89 | 28.0 | 0.84 | 29.9 | 0.91 | 25.8 | 0.78 |
| 11 | 26.5 | 0.67 | 27.6 | 0.84 | 29.8 | 0.64 | 30.0 | 0.75 | 30.1 | 0.88 | 27.0 | 0.82 |
| 12 | 29.5 | 0.78 | 30.9 | 0.86 | 32.9 | 0.92 | 33.5 | 0.87 | 33.9 | 0.91 | 28.9 | 0.79 |
| 13 | 27.2 | 0.75 | 28.5 | 0.82 | 30.8 | 0.86 | 30.4 | 0.85 | 31.0 | 0.89 | 27.7 | 0.79 |
| 14 | 24.5 | 0.69 | 25.5 | 0.78 | 27.9 | 0.84 | 26.3 | 0.81 | 28.1 | 0.89 | 24.8 | 0.75 |
| 15 | 25.7 | 0.62 | 25.9 | 0.68 | 30.0 | 0.85 | 27.6 | 0.74 | 28.8 | 0.82 | 25.7 | 0.66 |
| 16 | 27.2 | 0.63 | 27.9 | 0.71 | 30.1 | 0.81 | 29.1 | 0.74 | 29.8 | 0.81 | 27.5 | 0.68 |
| 17 | 31.3 | 0.85 | 31.9 | 0.87 | 33.2 | 0.90 | 33.5 | 0.88 | 34.0 | 0.91 | 31.3 | 0.86 |
| 18 | 32.9 | 0.75 | 33.7 | 0.78 | 32.7 | 0.77 | 35.4 | 0.81 | 35.3 | 0.84 | 33.2 | 0.76 |
| 19 | 24.9 | 0.76 | 26.0 | 0.83 | 28.9 | 0.90 | 29.0 | 0.89 | 28.9 | 0.93 | 25.2 | 0.80 |
| 20 | 32.8 | 0.90 | 32.9 | 0.92 | 32.7 | 0.93 | 35.5 | 0.93 | 35.7 | 0.94 | 31.3 | 0.92 |
| Avg. | 28.1 | 0.77 | 28.9 | 0.83 | 31.1 | 0.87 | 31.1 | 0.85 | 31.7 | 0.9 | 28.2 | 0.81 |

### 4.3. Comparison with State-of-the-Art Methods

To verity the advantage of the proposed method, we compare our proposed method with state-of-the-art inverse halftoning methods, where the halftone images are generated using Floyd Steinberg DD error diffusion method. Recently, three representative methods, named as SADCNN [17], GRL [20], PRL [18], achieved state-of-the-art results for inverse halftoning. Thus they are selected for comparison with our method. Due to the fact the GPL [20] code is not open, so we wrote it according to the paper. Maybe our experimental environments are different, but the average test result is not as high as in the original paper. Besides inverse halftoning methods based on deep learning, a typical traditional method based on dictionary learning, named as LLD [14] is also employed as a comparison method. The PSNR value and SSIM value are used to evaluate the performance of different inverse halftoning methods. We first select all test images in Figure 4 to compare different methods.

Figure 6 shows the comparison with state-of-the-art methods. As observed in Figure.6, our proposed method has a qualitative and quantitative superiority over the state-of-the-art methods. What's more, recently published paper in reference [39] proposed an inverse halftoning method via stationary wavelet domain. In this paper, six classic images are selected as the testing images, which are Koala, Cactus, Bear, Barbra, Shop and Pepper, corresponding to images in the 5th, 4th, 14th, 12th, 10th and 9th of Figure 4. The average PSNR of the five test images is 29.95 in reference [37] and 30.68 in our method. The results once again show our method has better performance than recently published solutions.

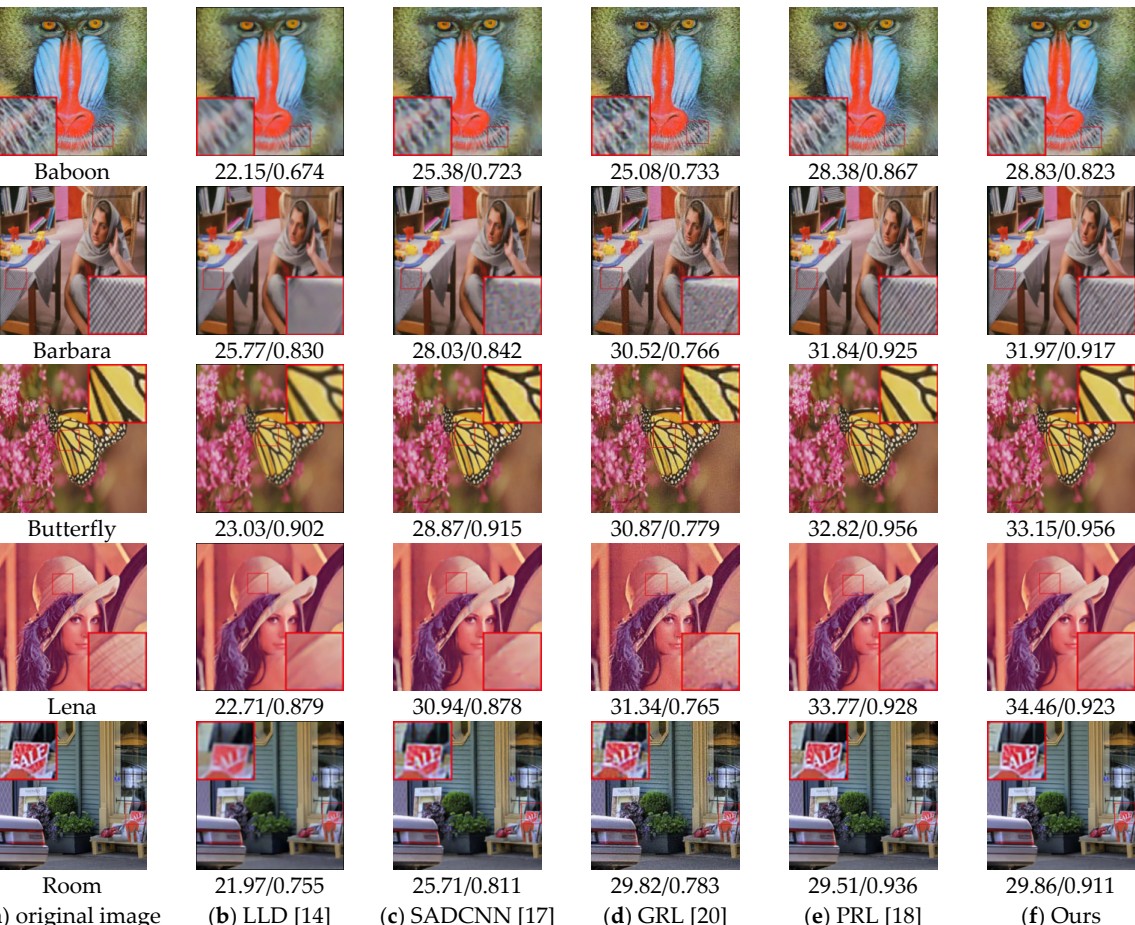

| | | | | | |
|---|---|---|---|---|---|
| Baboon | 22.15/0.674 | 25.38/0.723 | 25.08/0.733 | 28.38/0.867 | 28.83/0.823 |
| Barbara | 25.77/0.830 | 28.03/0.842 | 30.52/0.766 | 31.84/0.925 | 31.97/0.917 |
| Butterfly | 23.03/0.902 | 28.87/0.915 | 30.87/0.779 | 32.82/0.956 | 33.15/0.956 |
| Lena | 22.71/0.879 | 30.94/0.878 | 31.34/0.765 | 33.77/0.928 | 34.46/0.923 |
| Room | 21.97/0.755 | 25.71/0.811 | 29.82/0.783 | 29.51/0.936 | 29.86/0.911 |
| (**a**) original image | (**b**) LLD [14] | (**c**) SADCNN [17] | (**d**) GRL [20] | (**e**) PRL [18] | (**f**) Ours |

**Figure 6.** Comparison with state-of-the-art methods. For quantitative comparison, PSNR/SSIM are given.

To further show the advantages of our proposed method compared with the previous methods, we demonstrate some restored image details, lines and texture in Figures 7–9. In

Figure 7, the results of SADCNN [17] lack of texture details, but our approach describes the image details more accurately.

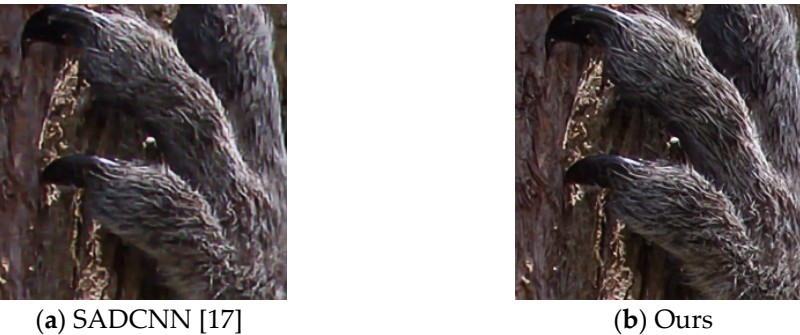

(**a**) SADCNN [17]  (**b**) Ours

**Figure 7.** Comparison with detail restoration between SADCNN and Ours.

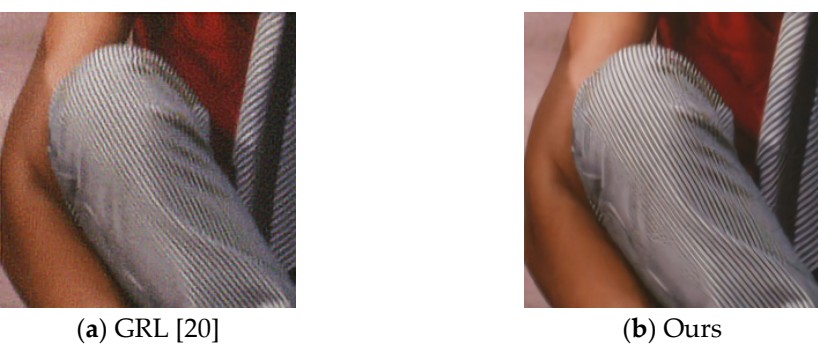

(**a**) GRL [20]  (**b**) Ours

**Figure 8.** Comparison with lines restoration between GRL and Ours.

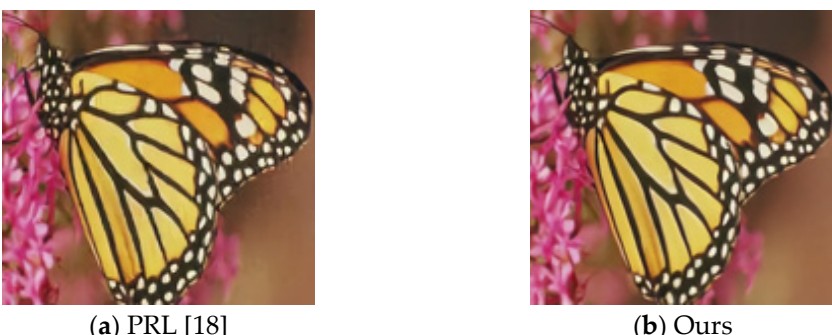

(**a**) PRL [18]  (**b**) Ours

**Figure 9.** Comparison with texture restoration between PRL and Ours.

In Figure 8, the restoration of GRL [20] cannot restore the lines of the image very well, but in our approach more lines and sharpness of the images are restored better. For the texture of restored Butterfly image as shown in Figure 9, there are some noise in the restored image by the PRL [18] method. However, the restored image by our method is smoother and more natural in texture restoration.

Besides, our approach could be applied to other datasets successfully. Table 4 gives the quantitative evaluation results, which demonstrates that methods based on deep learning are superior to traditional inverse halftoning methods. Moreover, our proposed method obtains the highest average PSNR performance compared with the other four methods and the SSIM value is similar to PRL [18]. Table 5 gives the results of comparison on PSNR/SSIM in 3600 test datasets obtained by different methods. From Table 5, we can see our proposed method achieves the best performance both in PSNR and SSIM compared with other methods.

**Table 4.** Comparison on PSNR/SSIM obtained by different methods.

| Test | LLD [14] | | SADCNN [17] | | GRL [20] | | PRL [18] | | Ours | |
|---|---|---|---|---|---|---|---|---|---|---|
| | PS NR | SS IM | PS NR | SS IM | PS NR | SS IM | PS NR | SS IM | PS NR | SS IM |
| 1 | 20.72 | 0.884 | 30.09 | 0.893 | 31.56 | 0.795 | 32.01 | 0.933 | 32.21 | 0.933 |
| 2 | 25.77 | 0.830 | 28.03 | 0.842 | 30.52 | 0.766 | 31.84 | 0.925 | 31.97 | 0.917 |
| 3 | 22.28 | 0.856 | 28.67 | 0.870 | 31.09 | 0.760 | 32.75 | 0.928 | 33.38 | 0.920 |
| 4 | 22.05 | 0.759 | 25.14 | 0.805 | 29.30 | 0.817 | 29.06 | 0.929 | 28.91 | 0.908 |
| 5 | 22.87 | 0.742 | 25.28 | 0.779 | 29.78 | 0.779 | 29.71 | 0.917 | 28.83 | 0.884 |
| 6 | 22.97 | 0.856 | 28.13 | 0.866 | 30.79 | 0.785 | 30.59 | 0.918 | 30.99 | 0.908 |
| 7 | 22.71 | 0.879 | 30.94 | 0.878 | 31.34 | 0.765 | 33.77 | 0.928 | 34.46 | 0.923 |
| 8 | 23.03 | 0.902 | 28.87 | 0.915 | 30.87 | 0.779 | 32.82 | 0.957 | 33.15 | 0.956 |
| 9 | 23.27 | 0.898 | 31.12 | 0.893 | 31.44 | 0.776 | 32.92 | 0.924 | 34.52 | 0.927 |
| 10 | 21.97 | 0.756 | 25.71 | 0.811 | 29.83 | 0.783 | 30.64 | 0.936 | 29.86 | 0.911 |
| 11 | 20.83 | 0.866 | 28.42 | 0.693 | 31.61 | 0.746 | 31.62 | 0.866 | 30.12 | 0.878 |
| 12 | 23.89 | 0.789 | 28.02 | 0.781 | 30.63 | 0.731 | 32.68 | 0.925 | 33.91 | 0.913 |
| 13 | 23.12 | 0.812 | 27.37 | 0.812 | 30.25 | 0.767 | 31.42 | 0.922 | 31.03 | 0.888 |
| 14 | 21.72 | 0.727 | 23.92 | 0.765 | 29.51 | 0.787 | 29.46 | 0.931 | 28.06 | 0.885 |
| 15 | 22.15 | 0.674 | 25.38 | 0.723 | 25.08 | 0.733 | 28.38 | 0.868 | 28.83 | 0.823 |
| 16 | 20.63 | 0.681 | 26.49 | 0.679 | 29.75 | 0.713 | 29.84 | 0.872 | 29.75 | 0.806 |
| 17 | 26.22 | 0.860 | 30.83 | 0.859 | 31.31 | 0.725 | 33.94 | 0.924 | 34.05 | 0.907 |
| 18 | 25.16 | 0.770 | 31.52 | 0.785 | 31.56 | 0.720 | 33.27 | 0.858 | 35.31 | 0.836 |
| 19 | 21.01 | 0.839 | 26.18 | 0.864 | 29.96 | 0.834 | 30.14 | 0.941 | 28.89 | 0.926 |
| 20 | 22.31 | 0.911 | 31.08 | 0.909 | 31.54 | 0.786 | 35.20 | 0.949 | 35.67 | 0.943 |
| Avg | 22.73 | 0.815 | 28.06 | 0.821 | 30.39 | 0.767 | 31.6 | 0.918 | 31.690 | 0.900 |

**Table 5.** Comparison on PSNR/SSIM in 3600 test datasets obtained by different methods.

| Methods | GRL [20] | SADCNN [17] | PRL [18] | Ours |
|---|---|---|---|---|
| PSNR | 27.529 | 28.375 | 31.328 | 31.447 |
| SSIM | 0.750 | 0.815 | 0.891 | 0.892 |

In the model training, the model loss and PSNR values were recorded in Figure 10, where PSNR is the average of 3600 test images.

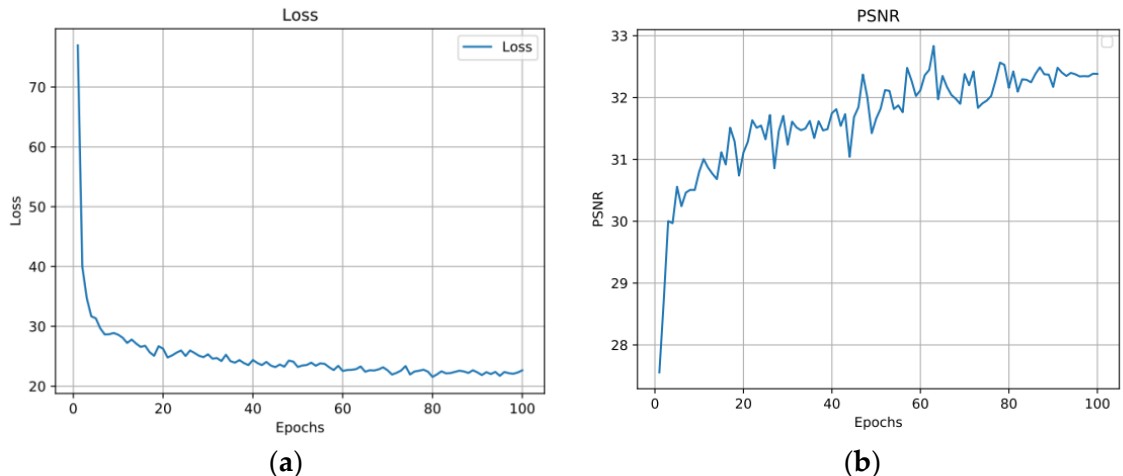

**Figure 10.** Results of model loss and PSNR changing with epochs. (**a**) Model loss; (**b**) Model PSNR.

*4.4. Ablative Study*

To evaluate the effect of the attention module and the parameter selection of loss function, we conducted an ablation experiment. All the evaluations are based on classical testing images in Figure 4.

Table 6 gives the effect of attention module. From Table 6, we can conclude that the attention module can improve the performance of the restored images. In addition, the restored images achieve highest PSNR and SSIM when RCAB is 16. Thus, the number of RCAB is selected 16 in our networks.

**Table 6.** The importance of RCAB for the inverse halftoning.

| Number of RCAB | PSNR | SSIM |
|---|---|---|
| RCAB-0 | 28.07 | 0.842 |
| RCAB-4s | 29.22 | 0.863 |
| RCAB-8s | 29.26 | 0.864 |
| RCAB-16s | 31.70 | 0.900 |
| RCAB-20s | 30.92 | 0.861 |

To obtain the suitable weight parameters for loss function, we do an experimental study as shown in Table 7. From Table 7, we can see the optimum values of $\lambda$ are $\lambda_1 = 0.1$, $\lambda_2 = 0.2, \lambda_3 = 0.3, \lambda_4 = 0.4$.

**Table 7.** The selection of loss function parameters.

| $\lambda$ | | | | PSNR | SSIM |
|---|---|---|---|---|---|
| $\lambda_1$ | $\lambda_2$ | $\lambda_3$ | $\lambda_4$ | | |
| 0.4 | 0.3 | 0.2 | 0.1 | 30.97 | 0.861 |
| 0.1 | 0.4 | 0.3 | 0.2 | 31.15 | 0.864 |
| 0.1 | 0.2 | 0.4 | 0.3 | 31.05 | 0.867 |
| 0.1 | 0.2 | 0.3 | 0.4 | 31.70 | 0.900 |

*4.5. Discussion*

In this article, we report a simple and effective network structure. Firstly, we use a stacked convolution layer to extract features effectively. Then we use a residual channel attention module to extract more effective features and guide the network to remove noise dot-patterns and restore image details. We realize the fusion of low-level features and high-level features through skip connection. Finally, the network is optimized by the multi-scale supervision loss, and the state-of-the-art experimental results are achieved.

The following points summarize our experimental results. (1) From the network structure design, the network is a lightweight network, simple and effective, easy to implement. (2) From the results of quantitative indicators PSNR and SSIM, this method achieves the state-of-the- art results compared with some previous methods. (3) From the qualitative analysis, the network has good restoration results, especially the restoration of the details, lines and texture, which is better than the previous methods. (4) From the point of view of the generality of the model, the previous method can only restore the single halftone image, our method can realize the restoration of many types of halftone images, and has generality.

## 5. Conclusions

We have proposed a novel deep convolutional neural network by fusing the attention mechanism for inverse halftoning. The proposed network first extracts main features and removes noisy-dot patterns by a convolution layer, and then reconstructs image by fusing the feature information and residual channel attention block. Such an attention block can help the network to focus on informative features. Thus, the restored image can have more fine details. Finally, we test on the classic datasets, the average PSNR is 31.70, and the

average SSIM is 0.9. The results of experiment show that our approach outperforms the state-of-the-art methods both in visual performance and in quantitative evaluation.

In the future research, the restoration of scanned halftone image is an urgent problem to be solved, where the most challenging problem is that there is no paired data. How to construct the inverse halftone method under the condition of unpaired data is an im-portant research direction. In addition, semi-supervised or unsupervised learning com-bined with our methods will be further studied in the future work.

**Author Contributions:** L.S. conceived the study and wrote the manuscript. E.Z. designed the research and conducted the experiment. M.L. proposed some valuable suggestions and revised the manuscript. All authors have read and agreed to the published version of the manuscript.

**Funding:** This research was funded by Scientific Research Program Foundation of Shaanxi Provincial Education Department, grant number 20JY053.

**Conflicts of Interest:** The authors declare no conflict of interest.

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
