# Peer review of "An Efficient Convolutional Neural Network Model Combined with Attention Mechanism for Inverse Halftoning"

_electronics, doi:10.3390/electronics10131574_

Round 1

Reviewer 1 Report

The authors proposed CNN model with attention mechanism for inverse halftoning. Overall paper is timely, and well presented. The authors explained the methodology well and evaluated using extensive simulation results. Please see my minor comments to improve the manuscript further.

  1. Add these latest deep learning papers as a reference.

https://doi.org/10.1109/TIP.2017.2736343

https://doi.org/10.1016/j.comcom.2020.08.017

  1. Add a comparative summary table of related work by highlighting paper aim, proposed method, pros and cons.
  2. Add detail experimental parameters in a tabular form.
  3. Add logical reasoning for the results attained.
  4. Add future work.

Reviewer 2 Report

The work presented in solid and the results suggest a good performance of the method proposed.

The presentation of the manuscript is good.

References are updated.

Reviewer 3 Report

 Inverse halftoning is considered one of the most trouble of accurate image restoration. In this manuscript, a proposed new deep learning model, convolutional neural network (CNN), integrated with fusion attention mechanism for inverse halftoning is applied. The experiment results show that the proposed approach beats the state-of-the-art methods not only in quantitative evaluation but also in visual performance. Hence, the idea is interesting and sounded, however, there are some issues that should be addressed by the authors:

1) The paper suffers from some language challenges. The paper should be proofread by a native speaker or a proofreading agent. Please, revise the readability and presentation well to be simple to the reader.

2) The Abstract section must be made much more impressive by highlighting your novelty.

3) It is suggested to check the citation reference of equation (1). In addition, check carefully all the abbreviation definitions in the whole manuscript. Further, please add different equations of the methodology of the proposed deep learning model in section 3.

4) I suggest in the start of section 2, the authors should add a new subsection that contains the conventional methods and much more impressive background by adding and citing the latest up-to-date references 2021 of the machine and deep learning techniques, e.g. supervised and unsupervised techniques (mdpi.com/1424-8220/21/4/1038, doi: 10.1109/ACCESS.2021.3083499, mdpi.com/1424-8220/21/2/487).

5) It is mandatory to add a discussion subsection before the conclusion section that describe more explanations of the proposed model and its superiority compared with the other state-of-art methods.

6) In the results section, you should draw the CNN training model accuracy and model loss versus epochs.

7) Please confirm that all of these figures (specifically, Figs. 4 and 5) are origin and didn't take from another paper or book, if not please cite its origin reference by copyright permission because it is a person's photos.

8) The conclusion section is weak; it should be rearranged, and numerical results should be added. According to the topic of the paper, the authors may propose some interesting problems as future work in the conclusion.

Round 2

Reviewer 3 Report

Thanks to the authors for these valuable modifications.